# Investigation of Camera-Free Eye-Tracking Glasses Compared to a Video-Based System

**DOI:** 10.3390/s23187753

**Published:** 2023-09-08

**Authors:** Abdullah Zafar, Claudia Martin Calderon, Anne Marie Yeboah, Kristine Dalton, Elizabeth Irving, Ewa Niechwiej-Szwedo

**Affiliations:** 1Department of Kinesiology & Health Sciences, University of Waterloo, Waterloo, ON N2L 3G1, Canada; a35zafar@uwaterloo.ca (A.Z.);; 2School of Optometry & Vision Science, University of Waterloo, Waterloo, ON N2L 3G1, Canada

**Keywords:** eye movements, validation, fixation, saccades, smooth pursuit

## Abstract

Technological advances in eye-tracking have resulted in lightweight, portable solutions that are capable of capturing eye movements beyond laboratory settings. Eye-tracking devices have typically relied on heavier, video-based systems to detect pupil and corneal reflections. Advances in mobile eye-tracking technology could facilitate research and its application in ecological settings; more traditional laboratory research methods are able to be modified and transferred to real-world scenarios. One recent technology, the AdHawk MindLink, introduced a novel camera-free system embedded in typical eyeglass frames. This paper evaluates the AdHawk MindLink by comparing the eye-tracking recordings with a research “gold standard”, the EyeLink II. By concurrently capturing data from both eyes, we compare the capability of each eye tracker to quantify metrics from fixation, saccade, and smooth pursuit tasks—typical elements in eye movement research—across a sample of 13 adults. The MindLink system was capable of capturing fixation stability within a radius of less than 0.5${}^{\circ }$, estimating horizontal saccade amplitudes with an accuracy of 0.04${}^{\circ }$± 2.3${}^{\circ }$, vertical saccade amplitudes with an accuracy of 0.32${}^{\circ }$± 2.3${}^{\circ }$, and smooth pursuit speeds with an accuracy of 0.5 to 3$\frac{{}^{\circ }}{s}$, depending on the pursuit speed. While the performance of the MindLink system in measuring fixation stability, saccade amplitude, and smooth pursuit eye movements were slightly inferior to the video-based system, MindLink provides sufficient gaze-tracking capabilities for dynamic settings and experiments.

## 1. Introduction

Eye movements have been widely used to probe sensorimotor and cognitive mechanisms giving us deeper insight into human behavior and information processing [1,2], for example, attention [3,4,5], working memory [6,7], decision-making [8], and learning [9,10]. The neural control of eye movements engages multiple brain networks, thus, eye-tracking methodologies have been used to understand development and aging [11,12,13], to help with medical diagnostics [14,15], and to monitor disease progression [16]. More recently, eye tracking has been applied to study complex behaviors in natural environments [17], for example, navigation [18,19], occupational performance [20,21], and education [22,23].

Eye-tracking methodologies have evolved greatly over time mainly due to technological and computational advances. Cognolato, Atzori, and Müller (2018) outlined seven different eye-tracking methods: (1) electrooculogram (EOG), (2) electromagnetic methods (i.e., scleral search coils), (3) contact lenses, (4) limbus/iris–sclera boundary video-oculography, (5) pupil video-oculography, (6) pupil and corneal reflections video-oculography, and (7) dual Purkinje image corneal reflection video-oculography (see [24] for a complete review of each technique). EOG and search coils are two camera-free eye-tracking approaches currently available. Search coils provide the most accurate and precise recordings; however, this approach is the most invasive and restrictive because a copper wire coil is embedded in a contact lens and the recording must be completed while the participant is sitting in an oscillating magnetic field. The voltage recorded by the coils represents the orientation of the eyes in the magnetic field. On the other hand, EOG is less invasive because the signal, generated by changes in the electrical potential between the cornea and the retina when the eyes move, is recorded by electrodes placed on the outer canthi of the eyes. The main limitation of the EOG approach is lower accuracy in comparison to other systems [25].

The most frequently used technique with human participants is video-oculography, which is a non-invasive, video-based approach where videos of the pupil and corneal reflection are recorded and processed to estimate eye position. Such systems are traditionally expensive and heavy from the weight of the cameras and the corresponding frames needed to support the camera. Moreover, in order to achieve a high level of precision, these systems require users to restrict head motion and, therefore, lack portability or utility in more dynamic scenarios beyond constrained laboratory-based studies. In recent years, there has been an emergence of more mobile and portable gaze trackers that have allowed researchers to move into more dynamic and natural environments. However, these systems still rely on video-based technology, which limits the flexibility and processing power capabilities of these smaller (more portable) devices.

A new camera-free eye-tracking technology has emerged, which uses compact microelectromechanical systems (MEMSs) to scan the eyes with low-powered beams. MEMSs allow gaze estimation without relying on the more resource-expensive (computationally and power-consumption-wise) image processing of video camera feeds. The MindLink system by AdHawk Microsystems is an example of such a camera-free system, where the entire MEMS-based tracking system is embedded into eyeglass frames, facilitating portability and utility beyond highly constrained laboratory experiments. While there is great potential for such advancements in technology to stimulate eye-movement research studies in ecological settings, a rigorous validation against the current research grade, lab-based “gold standard” must be established.

To date, the only published evaluation of the signal quality from the MindLink system involved a comparison with the EyeLink II eye tracker, using the simultaneous data capture of a single eye per system [26]. The signal quality was mainly evaluated as the spatial accuracy and precision of the measured gaze position to a stimulus target. However, additional metrics are frequently used in research studies to characterize eye movements, such as saccade amplitude ([27]), fixation stability, peak velocity, and smooth pursuit velocity gain ([28]). As such, the aim of the current work was to compare the paired eye movement recordings simultaneously captured from the EyeLink II and MindLink eye trackers in order to present a detailed analysis of the validity and capability of the MindLink system in measuring fixations, saccades, and smooth pursuit eye movements.

## 2. Materials and Methods

### 2.1. Eye-Tracking Devices

EyeLink II is a head-mounted, video-based eye tracker (SR Research Ltd., Ottawa, ON, Canada) capable of recording either monocular or binocular gaze position using pupil-only or pupil–corneal reflection tracking. Two eye-tracking cameras (one for each eye) are attached to an adjustable headband, which fits on the head of the user.

The AdHawk MindLink is a camera-free eye tracker that uses a microelectromechanical system (MEMS) to detect the binocular gaze position at a sampling frequency of up to 500 Hz. The MEMS is fit into an eyeglass frame and comprises an infrared scanner located behind each nosepiece along with five infrared sensors distributed around the inside of the eyeglass rim. The scanner emits a low-powered infrared beam across the eye, and the resulting corneal reflections are detected by the infrared sensors, which are then used to determine pupil position.

In the current study, the sampling frequency for both eye trackers was 250 Hz. Eye trackers were calibrated separately using a nine-point calibration grid following the instructions from the user’s guide for each device (Figure 1). Validation was also performed and accepted when the tracking error was less than 1 degree.

#### EyeLink-MindLink Signal Alignment

Temporal synchronization between the eye trackers was achieved using a TTL pulse sent from the Experiment Builder, which controlled the stimulus presentation. A TTL pulse was sent to an Arduino Uno at the start of each trial and each pulse was time-stamped using custom Python scripts on the PC system running the MindLink system. The time-stamp was used to split the MindLink recordings into trials corresponding to the EyeLink sample reports exported from the Experiment Builder. Gaze positions between eye trackers were then further aligned for each trial using the maximum of a normalized cross-correlation function between the gaze positions from each eye tracker. Trials where the cross-correlation between signals was less than 0.5 were excluded for poor tracking (13.5% of all trials).

### 2.2. Participants

Thirteen participants were recruited from the University of Waterloo, Canada. All participants (4 males/9 females, mean age 23.5 ± 3.4) had normal or corrected-to-normal vision. They reported no neurological or musculoskeletal disorders. The study was reviewed and received ethics clearance through the University of Waterloo’s Office of Research Ethics Committee (ORE no. 43620). Written consent was obtained from all participants.

### 2.3. Stimulus and Tasks

All visual stimuli were generated using the Experiment Builder (version 2.3.38, SR Research), and presented on a 19 inch CRT monitor (Samsung SyncMaster; resolution 1024 × 768, 85 Hz refresh rate). Each participant was seated comfortably with their head stabilized in a chin rest. Participants completed all five tasks described below in a pseudo-randomized order (see Figure 2 for an example of the experimental task order).

#### 2.3.1. Fixation Task

Participants completed five fixation trials. The fixation target was a black dot, measuring 0.25${}^{\circ }$ of the visual angle, presented on a white background at the center of a monitor positioned 100 cm in front of the participant. During each trial, participants were asked to look at the dot for a duration of 5 s.

#### 2.3.2. Saccade Tasks

Participants completed horizontal and vertical saccade tasks. For both tasks, participants were seated 80 cm away from the monitor. At the beginning of each trial, a white fixation cross was presented on a black background for a duration ranging from 1000 to 1500 ms. The saccade target was a white dot (diameter 0.25${}^{\circ }$), which appeared to follow the fixation cross; this target was then followed by a second target presented in the opposite hemifield. Participants were instructed to start the next trial at their discretion.

Horizontal saccades: For the horizontal saccade task, targets were presented to the left or the right of the fixation (target amplitude range 3–12${}^{\circ }$ (see Table 1)). Participants completed 175 trials.

Vertical saccades: For the vertical saccade task, targets were presented above or below the fixation (target amplitude range 0.5–17${}^{\circ }$ (see Table 1)). Participants completed 60 trials.

#### 2.3.3. Smooth Pursuit Tasks

Participants completed a horizontal and vertical smooth pursuit task. Participants started each trial by pressing a button on the keyboard at their discretion. During each trial, participants were asked to fixate on a white fixation cross presented on a black background in the center of the monitor located 80 cm in front of them. The fixation cross disappeared 1000 ms after the trial initiation, a white target dot appeared and began oscillating at one of the three pseudo-randomized speeds (see Table 1). Participants were asked to follow the dot (i.e., keep their eyes on the moving target) as it moved across the screen.

Horizontal smooth pursuit: For the horizontal smooth pursuit task, the white target dot appeared on the left or right of the fixation cross before oscillating (see Table 1 for the oscillating speeds). Participants completed 30 horizontal smooth pursuit trials.

Vertical smooth pursuit: For the vertical smooth pursuit task, the white target dot appeared above or below the fixation cross before oscillating (see Table 1 for the oscillating speeds). Participants completed 30 vertical smooth pursuit trials.

### 2.4. Analysis

#### 2.4.1. Conversion to Degrees of the Visual Angle

The data from each system were converted into degrees of the visual angle either from display screen pixels (EyeLink) or a gaze vector (MindLink).

For the pixel-to-degree conversion, the following functions were used:$$x_{\text{degrees-EL}}=\arctan\left(\frac{w_{\mathrm{cm}}\cdot \left(\frac{x_{\mathrm{px}}}{w_{\mathrm{px}}}-0.5\right)}{d_{\mathrm{cm}}}\right)$$
$$y_{\text{degrees-EL}}=\arctan\left(\frac{h_{\mathrm{cm}}\cdot \left(\frac{y_{\mathrm{px}}}{h_{\mathrm{px}}}-0.5\right)}{d_{\mathrm{cm}}}\right)$$

Here, $h_{\mathrm{cm}},w_{\mathrm{cm}}$ are the height and width of the display screen in centimeters, respectively (with constant values of $h_{\mathrm{cm}}$ = 35.2 cm and $w_{\mathrm{cm}}$ = 26.4 cm). Similarly, $h_{\mathrm{px}},w_{\mathrm{px}}$ are the height and width of the display screen in pixels (with $h_{\mathrm{px}}$ = 1024 and $w_{\mathrm{px}}$ = 768). The parameter $d_{\mathrm{cm}}$ represents the distance from the participant to the display screen (constant at $d_{\mathrm{cm}}$ = 80 cm). Finally, $x_{\mathrm{px}},y_{\mathrm{px}}$ are the raw gaze position coordinates in display screen pixels obtained from the EyeLink sample reports.

For the gaze vector-to-degree conversion, the following MATLAB functions were used:$$x_{\text{degrees-ML}}=\arctan\left(\frac{v_{x}}{\sqrt{v_{y}^{2}+v_{z}^{2}}}\right)$$
$$y_{\text{degrees-ML}}=\arctan\left(\frac{v_{y}}{-v_{z}}\right)$$

Here, $v_{x}$, $v_{y}$ are the horizontal and vertical components of the instantaneous gaze vector reported by the MindLink system, respectively. The gaze vector is a unit vector of the form $v_{\mathrm{gaze}}=\left(v_{x},v_{y},v_{z}\right)$ with the origin set at the center of the pupil, with the x-axis oriented toward the right, the y-axis oriented upward, and the z-axis oriented backward from the participant’s point of view.

The average eye position (mean of the right eye and left eye) was used to extract eye movement features for the comparison between the two eye trackers, while individual eye streams were used to compare the right and left eye conjugacy within each tracker.

#### 2.4.2. Fixation Stability

Fixation stability was assessed by calculating the standard deviation of the horizontal (x) and vertical (y) positions of the left and right eyes, which were then used to calculate a log-transformed bivariate contour ellipse area (log${}_{10}$BCEA). Fixation stability was assessed during a 2000 ms fixation interval, which did not contain blinks or other noise artifacts. The following formula was used to calculate log${}_{10}$BCEA [29]: (1)$$\log_{10}\mathrm{BCEA}=\log_{10}\left(\pi \chi^{2}\sigma_{x}\sigma_{y}\sqrt{1-\rho^{2}}\right)$$
where $\chi^{2}=2.291$ is the chi-square value (2 degrees of freedom) corresponding to a probability value of P=0.682 (±1 standard deviation), $\sigma_{x}$ is the standard deviation of the horizontal eye position, $\sigma_{y}$ is the standard deviation of the vertical eye position, and ρ is the Pearson correlation coefficient.

#### 2.4.3. Saccades

Saccades were automatically detected using amplitude, velocity, and acceleration thresholds of 0.01${}^{\circ }$, 30 $\frac{{}^{\circ }}{s}$, and 800 $\frac{{}^{\circ }}{s^{2}}$, respectively [30]. For each saccade in the EyeLink data, a corresponding saccade in the MindLink data was selected if it was within a 100 ms window of the start of the EyeLink saccade. A total of 12.7% of trials were excluded (14.2% for horizontal saccades, 12.1% for vertical saccades) due to poor tracking where the cross-correlation was less than 0.5 between the signals for each eye tracker.

The main outcome measures to evaluate saccades between the two systems were the amplitude and the asymptotic peak velocity extracted from the main sequence. Amplitude was defined as the change in position (horizontal or vertical) in degrees for the detected saccade.

The main sequence was obtained by plotting the amplitude (${}^{\circ }$, abscissa) and peak saccade velocity ($\frac{{}^{\circ }}{s}$, ordinate) for all saccades for each participant. Curves were then fit to the data using: (2)$$V=V_{\max}\cdot \left[1-\exp(-c\cdot A)\right]$$
where *A* and *V* are the saccade amplitude and velocity data, respectively, and the fit parameters are *c* and $V_{\max}$—the asymptotic peak velocity. Curve fitting is achieved with SciPy (Version 1.11.1) using a trust region reflective algorithm (method = ‘trf’), seed parameters of (*c* = 1,$V_{\max}$ = 500), and a Cauchy loss function (loss = ‘cauchy’). $V_{\max}$ is extracted for both systems separately for all participants.

Finally, the saccade conjugacy between the left and right eye is evaluated by calculating amplitude differences between the left and right eye (right eye saccade amplitude minus the left eye saccade amplitude) for each participant. The participants’ amplitude difference average was calculated and statistically compared against a value of zero using a one-sample *t*-test for each eye tracker.

#### 2.4.4. Smooth Pursuit

In order to compare the smooth pursuit between eye trackers, the gaze position from each eye tracker was first temporally aligned with the target position by shifting the gaze signal according to the maximum cross-correlation function between the gaze and target. Saccades were then automatically detected (as in Section 2.4.3) and removed from the trial.

The main outcome measure used to compare tracking quality during the smooth pursuit tasks between EyeLink and MindLink was the velocity gain, which is defined as the ratio of eye velocity to target velocity. Velocity gain was computed for the linear portions of the sinusoidal pursuit trials. For the two faster target frequencies (0.1 Hz and 0.2 Hz), the linear motion occurred over a 1000 ms window centered at the peak target velocity. For the slowest target frequencies (0.01 Hz), the entire trial was approximated as linear motion, and a 1000 ms window at the start of the trial was used. Smooth pursuit data were subjected to a linear fit using the MATLAB function: fit(*t*, *r*, ‘poly1’), where *t* is the timestamp for each sample and *r* is the gaze position. The first coefficient of the linear fit (i.e., the slope of the function) was used as an estimate of smooth pursuit velocity. A corresponding linear fit of the target position data was performed to estimate target velocity. A single measure of velocity gain was then calculated as the velocity from the eye tracker divided by the velocity of the target motion.

Trials were removed from the analysis if the maximum cross-correlation between the gaze and target was less than 0.5 or when less than 200 ms of data remained available for the linear fit after saccade removal. In total, 9.8% of trials were excluded (6.9% for horizontal pursuit, 12.6% for vertical pursuit) due to poor tracking, where the cross-correlation was less than 0.5 between the signals from each eye tracker.

The conjugacy between the left and right eyes during the smooth pursuit was calculated as the average velocity gain difference between eyes (right eye velocity gain minus left eye velocity gain). The mean conjugacy across all trials for each participant was then calculated for each eye tracker at each speed level and compared against a target value of zero using a two-way repeated-measure analysis of variance (ANOVA) with the eye tracker and speed factors.

## 3. Results

Validity (bias) and limits of agreement between the measures obtained from the two eye trackers were examined using a Bland–Altman test.

### 3.1. Fixation Stability

Figure 3 shows typical fixation eye position profiles recorded with EyeLink II and AdHawk MindLink; Table 2 outlines summary statistics of fixation stability measures for all participants for both eye trackers. Fixation stability was examined by comparing the standard deviation (SD) of the horizontal and vertical eye positions between devices. Figure 4 shows Bland–Altman plots, illustrating the distribution of differences between the two eye-tracking systems in the standard deviation of the horizontal and vertical eye positions across all trials for each participant. The mean difference and 95% confidence interval between the two trackers (EyeLink and MindLink) was −0.08${}^{\circ }$ [−0.32, 0.17] for the horizontal standard deviation and −0.08${}^{\circ }$ [−0.37, 0.21] for the vertical standard deviation. The mean differences across participants are shown in Figure 5.

A Bland–Altman plot for the global measure of fixation stability, log${}_{10}$BCEA, is shown in Figure 6. The differences between the devices for each participant are shown in Figure 7. The mean difference and 95% confidence interval between the two trackers (EyeLink and MindLink) was −0.29 degrees${}^{2}$ [−0.98, 0.4] for fixation log${}_{10}$BCEA.

### 3.2. Saccades

Figure 8 shows typical horizontal and vertical saccade eye position profiles recorded with EyeLink II and AdHawk MindLink.

#### 3.2.1. Horizontal Saccades

Horizontal saccades were characterized by comparing the amplitude measured by each device. Figure 9 shows a Bland–Altman plot illustrating the horizontal saccade amplitude agreement between the devices for both eyes. The mean difference and 95% confidence interval between the two trackers (EyeLink and MindLink) was 0.04${}^{\circ }$ [−4.56, 4.63] for the horizontal saccade amplitude. For each participant, the measurement differences between devices are shown in Figure 10.

A horizontal main sequence plot using data from all participants and trials is shown in Figure 11. The dashed lines represent the calculated asymptotic maximum velocity for each system. The Bland–Altman plot in Figure 12 demonstrates the difference in detected horizontal maximum velocity between the two systems for each participant, where the mean difference and 95% confidence interval between the two trackers (EyeLink and MindLink) was −10.70 $\frac{{}^{\circ }}{s}$ [−204.57, 183.17]. For each participant, the differences in detected horizontal maximum velocity values between the devices are shown in Figure 13.

#### 3.2.2. Vertical Saccades

Vertical saccade characterization was examined by comparing the vertical saccade amplitude measured by each device. Figure 14 shows a Bland–Altman plot illustrating the vertical saccade amplitude agreement between the devices for both eyes for the first target saccade. The mean difference and 95% confidence interval between the two trackers (EyeLink and MindLink) was −0.32${}^{\circ }$ [−4.84, 4.20] for the vertical standard deviation. For each participant, the measurement differences between devices are shown in Figure 15.

A vertical main sequence plot using data for all participants and trials is shown in Figure 16. The dashed lines represent the calculated asymptotic maximum velocity for each system. The Bland–Altman plot in Figure 17 demonstrates the difference in detected vertical maximum velocity between the two systems for each participant, where the mean difference and 95% confidence interval between the two trackers (EyeLink and MindLink) was 204.21 $\frac{{}^{\circ }}{s}$ [−266.22, 674.64]. For each participant, differences in detected vertical maximum velocity values between devices are shown in Figure 18.

#### 3.2.3. Saccade Conjugacy

Saccade conjugacy was tested for both eye trackers by performing a one-sample t-test on the saccade amplitude difference between eyes (right–left) for each eye tracker. For both vertical and horizontal saccades, there was a significant deviation of conjugacy from the ideal zeros in both eye trackers (*p* < 0.01). The summary statistics and *t*-test results for horizontal and vertical saccade amplitude differences are outlined in Table 3.

### 3.3. Smooth Pursuit

Figure 19 shows typical smooth pursuit eye position profiles recorded with EyeLink II and AdHawk MindLink, and Table 4 outlines the summary statistics of horizontal and vertical smooth pursuit measures from all participants for both eye trackers.

#### 3.3.1. Horizontal Smooth Pursuit

Horizontal smooth pursuit characterization was examined by comparing the horizontal velocity gain measured by each device. Figure 20 shows the Bland–Altman plots, illustrating the horizontal velocity gain agreement between the devices for both eyes for each pursuit speed. The mean horizontal gain differences and 95% confidence intervals between the two trackers (EyeLink and MindLink) were 0.18 [−1.71, 2.07], 0.07 [−0.28, 0.42], and 0.06 [−0.25, 0.37] for horizontal pursuit speeds of 0.5 $\frac{{}^{\circ }}{s}$, 5 $\frac{{}^{\circ }}{s}$, and 10 $\frac{{}^{\circ }}{s}$, respectively. For each participant, the horizontal velocity gain differences between devices are shown in Figure 21.

For each participant, the horizontal velocity measurement differences between devices are shown in Figure 22. The mean horizontal velocity differences and 95% confidence intervals between the two trackers (EyeLink and MindLink) were −0.02 $\frac{{}^{\circ }}{s}$ [−1.11, 1.07], 0.09 $\frac{{}^{\circ }}{s}$ [−1.78, 1.97], and 0.04 $\frac{{}^{\circ }}{s}$ [−3.55, 3.63] for horizontal pursuit speeds of 0.5 $\frac{{}^{\circ }}{s}$, 5 $\frac{{}^{\circ }}{s}$, and 10 $\frac{{}^{\circ }}{s}$, respectively.

#### 3.3.2. Vertical Smooth Pursuit

Vertical smooth pursuit characterization was examined by comparing the vertical velocity gain measured by each device. Figure 23 shows Bland–Altman plots, illustrating the vertical velocity gain agreement between the devices for both eyes for each pursuit speed. The mean vertical gain differences and 95% confidence intervals between the two trackers (EyeLink and MindLink) were 0.09 [−2.16, 2.35], 0.06 [−0.53, 0.64], and 0.08 [−0.24, 0.39] for vertical pursuit speeds of 0.5 $\frac{{}^{\circ }}{s}$, 5 $\frac{{}^{\circ }}{s}$, and 10 $\frac{{}^{\circ }}{s}$, respectively. For each participant, vertical velocity gain differences between devices are shown in Figure 24.

For each participant, vertical velocity measurement differences between devices are shown in Figure 25. The mean vertical velocity differences and 95% confidence intervals between the two trackers (EyeLink and MindLink) were −0.07 $\frac{{}^{\circ }}{s}$ [−1.34, 1.20], −0.21 $\frac{{}^{\circ }}{s}$ [−3.16, 2.74], and 0.04 $\frac{{}^{\circ }}{s}$ [−3.76, 3.83] for vertical pursuit speeds of 0.5 $\frac{{}^{\circ }}{s}$, 5 $\frac{{}^{\circ }}{s}$, and 10 $\frac{{}^{\circ }}{s}$, respectively.

#### 3.3.3. Smooth Pursuit Conjugacy

Smooth pursuit conjugacy was tested for both eye trackers by performing a two-way repeated-measure ANOVA, where the dependent variable was the gain difference between eyes (right–left), with pursuit speed (0.5 $\frac{{}^{\circ }}{s}$, 5 $\frac{{}^{\circ }}{s}$, and 10 $\frac{{}^{\circ }}{s}$), and eye trackers (EyeLink, MindLink, or ‘Ideal’—gain difference set to 0). The ANOVA results showed no significant gain differences (p>0.05) between eyes for either eye tracker, in both the horizontal smooth pursuit (Table 5) and vertical smooth pursuit (Table 6).

## 4. Discussion

The aim of this study was to evaluate the ability of the MindLink eye tracker to capture eye movement characteristics frequently used in eye-tracking studies. The MindLink eye tracker is embedded in the eyeglass frame and uses novel camera-free technology to record eye positions. This technology may allow researchers to move out of the lab and into more ecological environments. As such, we compared the capabilities of the MindLink system to the traditional video-based EyeLink II system. Assessing the capabilities and limitations of a novel mobile system such as MindLink is necessary in order to understand the proper use cases of such a system. Previously, the only study assessing MindLink examined the accuracy and precision of the signal quality compared to an EyeLink eye tracker [26]. Here, we examined eye movement metrics frequently reported in eye-tracking studies.

### 4.1. Fixation

The Bland–Altman analysis for fixation stability (Figure 4 and Figure 6) showed greater variability in the MindLink recordings compared to the EyeLink recordings. These results are consistent with a previous study [26], where MindLink demonstrated poorer spatial precision. Notably, the difference in the fixation eye position between EyeLink and MindLink was less than 0.5${}^{\circ }$ (Table 2). This level of precision is most likely sufficient for studies conducted in dynamic settings, such as sports, where fixation differences greater than 0.5${}^{\circ }$ have been observed between expertise levels [31]. However, when the expected effect size between groups or conditions is less than 0.5${}^{\circ }$, the MindLink may not be a suitable device for detecting such differences.

### 4.2. Saccades

Saccade amplitude measurements from MindLink demonstrated high variability of agreement with the EyeLink system when considering individual participants (Figure 10 and Figure 15). The agreement lacked consistency because MindLink both under- and over-estimated saccade amplitudes and velocity gains within and across participants. In dynamic settings, such as driving, large amplitude saccades are important for performance and are defined with amplitudes greater than 10${}^{\circ }$ [27]. The MindLink amplitude measurement error relative to EyeLink’s had a standard deviation of around 2.5${}^{\circ }$ (Figure 9 and Figure 14), demonstrating MindLink’s ability to measure and discriminate large-scale eye movements relevant to real-world applications. Regarding the measurement of peak saccade velocity, video-based systems have previously demonstrated measurement inaccuracies compared to scleral search coil methods [32], and as such, it is difficult to know whether it is EyeLink or MindLink that is contributing more to the discrepancies in peak velocity estimation.

### 4.3. Smooth Pursuit

Smooth pursuit velocity gain differences between the eye trackers were dependent on the pursuit speed, with higher pursuit speeds demonstrating more agreement in velocity gain (Figure 20 and Figure 23). The high discrepancy in gain for the slower velocities was likely due to a lower limit on the pursuit velocity estimation accuracy of about 0.5 $\frac{{}^{\circ }}{s}$ for horizontal pursuit and up to 3 $\frac{{}^{\circ }}{s}$ for vertical pursuit, as apparent from Figure 22 and Figure 25 and Table 4. As such, the measurement of the slowest pursuit speed of 0.5 $\frac{{}^{\circ }}{s}$ could have an error of 0.5 $\frac{{}^{\circ }}{s}$, resulting in a large velocity gain estimation. However, the pursuit of targets moving at higher velocities (≥10$\frac{{}^{\circ }}{s}$), which is common in dynamic scenes [33,34,35], can be captured relatively accurately by MindLink. Finally, the MindLink performance is comparable to another mobile eye tracker, the Pupil Labs glasses, which presented a pursuit velocity estimation error of about 2.4 $\frac{{}^{\circ }}{s}$ relative to an EyeLink video-based system [36].

### 4.4. Limitations

This study offers the first investigation of common eye movement metrics recorded using a camera-free eye-tracking system. The new system was compared with a research-grade video-based eye tracker during a simultaneous recording. While this allowed us to compare the same eye movements recorded with both systems, it also introduced the possibility of interference between devices. To address these issues, pilot testing was conducted prior to the experiment where eye movements were recorded separately by each eye tracker, and the recordings were compared to assess the signals from the single vs. simultaneous recordings. Results from the pilot study did not reveal any significant interference; nonetheless, signal interference between the devices could have had some small influence on our results. Another limitation of the study is the limited number of tasks used (i.e., fixation, saccade, and smooth pursuit). These tasks have been used widely in eye movement studies; however, the capability of the MindLink system remains to be examined in other tasks where participants are not constrained by a chin rest and the stimulus is presented in more ecological conditions rather than a computer monitor. Finally, the application of MEMS technology to eye tracking is quite novel [37]. While the current tracking algorithms provide less accurate recordings when compared to the EyeLink eye tracker, it is likely that as the technology matures the tracking algorithms will also improve.

## 5. Conclusions

The AdHawk MindLink is a camera-free eye-tracking system, which presents many advantages over traditional video-based eye-tracking systems; these include a lighter weight, a more compact design, and greater portability outside of constrained postures and settings. While the performance of an effective video-based system, such as EyeLink II, is superior at measuring eye movement details, the MindLink stands out for its smaller footprint, and it maintains an adequate level of accuracy that is suitable for dynamic environments. Further work should explore the factors contributing to the accuracy variability observed among participants, as well as the performance of MEMS technology used by MindLink (involving infrared light) in natural light settings.

## Figures and Tables

**Figure 1 sensors-23-07753-f001:**
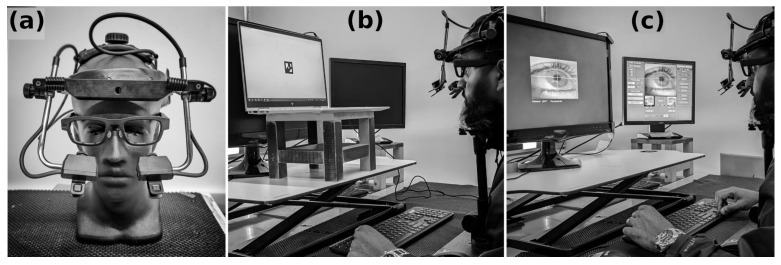
Participant instrumentation and calibration: (**a**) The EyeLink headset was worn on top of the MindLink frames, with the EyeLink cameras viewing the eyes through the empty frame front, (**b**) the MindLink system was calibrated on its collection PC laptop, (**c**) the EyeLink system was calibrated using the pupil-only mode on the stimulus display monitor. For both calibrations, participants’ heads were fixed in a chin-rest.

**Figure 2 sensors-23-07753-f002:**
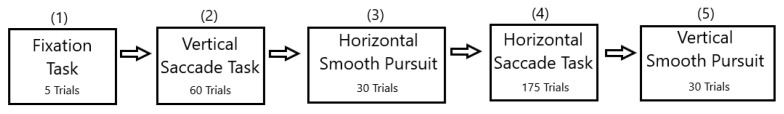
Example of one possible ordering of the pseudo-randomized experimental task procedure.

**Figure 3 sensors-23-07753-f003:**
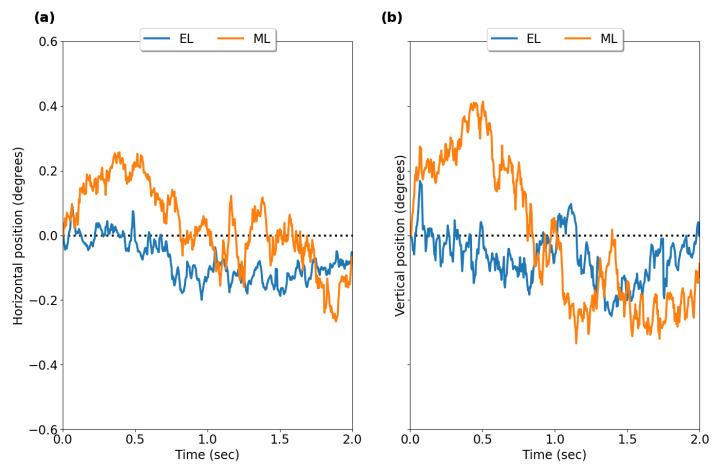
Raw fixation profiles. Exemplar single trial fixation profiles from average (left and right) eye position data recorded from EyeLink II (EL) and AdHawk MindLink (ML): (**a**) horizontal position, (**b**) vertical position.

**Figure 4 sensors-23-07753-f004:**
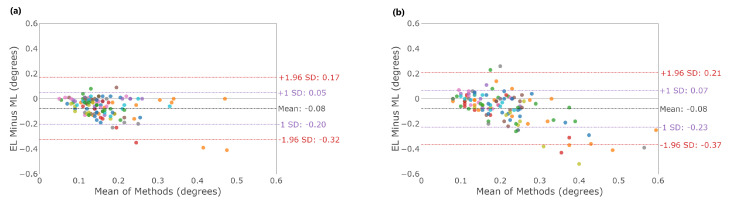
Bland–Altman plot in the fixation eye position standard deviation; (**a**) horizontal standard deviation; (**b**) vertical standard deviation. Colors distinguish individual participants.

**Figure 5 sensors-23-07753-f005:**
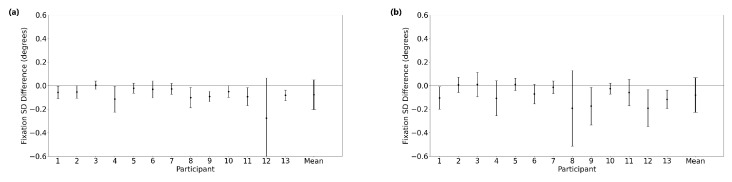
Mean ± SD difference (EyeLink and MindLink) of the fixation eye position. (**a**) Horizontal standard deviation. (**b**) Vertical standard deviation. Mean ± SD for each participant as well as the mean ± SD difference averaged across all participants and trials shown.

**Figure 6 sensors-23-07753-f006:**
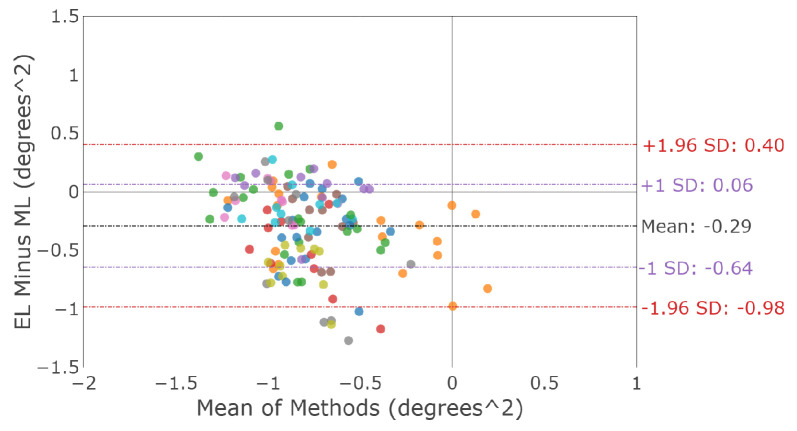
Bland–Altman plot of the fixation eye position bivariate contour ellipsoid area (log-transformed). Colors distinguish individual participants.

**Figure 7 sensors-23-07753-f007:**
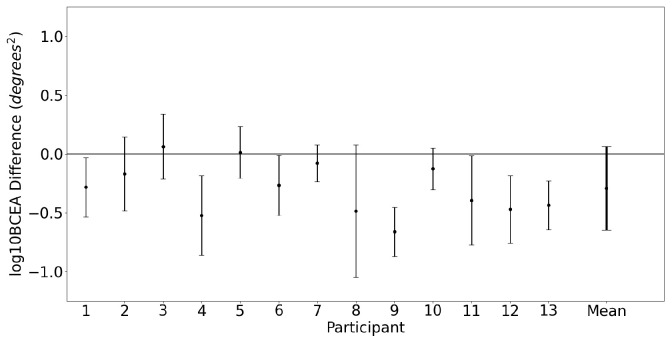
Mean ± SD difference (EyeLink and MindLink) in the fixation eye position log${}_{10}$BCEA for each participant, as well as the mean ± SD difference averaged across all participants and trials.

**Figure 8 sensors-23-07753-f008:**
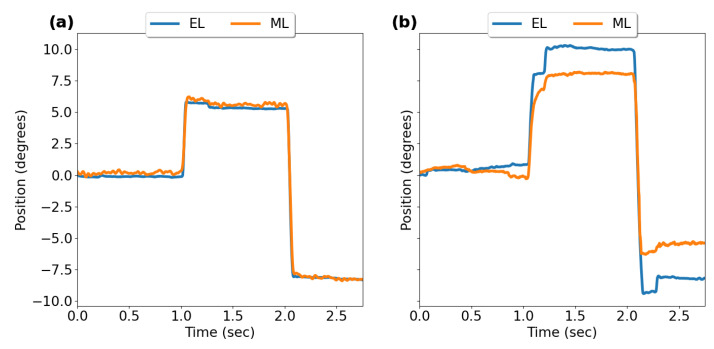
Saccade eye position profiles from average gaze position data recorded from EyeLink II (EL) and AdHawk MindLink (ML): (**a**) horizontal saccade, (**b**) vertical saccade.

**Figure 9 sensors-23-07753-f009:**
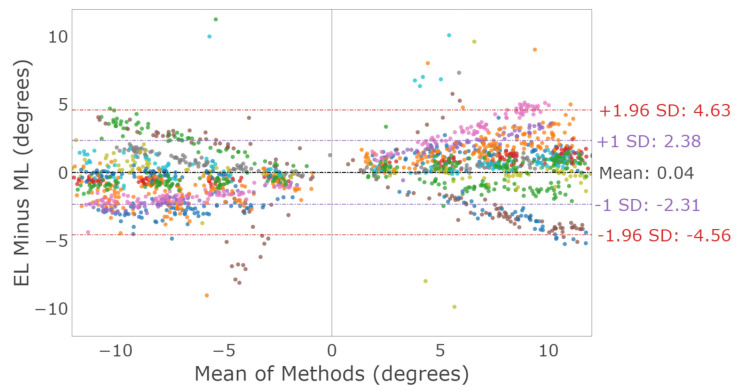
Bland–Altman plot of the horizontal saccade amplitude between EyeLink and MindLink. Colors distinguish individual participants.

**Figure 10 sensors-23-07753-f010:**
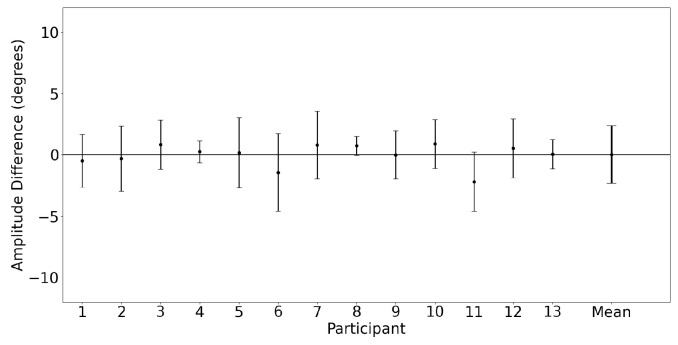
Mean ± SD difference (EyeLink and MindLink) of the horizontal saccade amplitude for each participant, as well as the mean ± SD difference averaged across all participants and trials.

**Figure 11 sensors-23-07753-f011:**
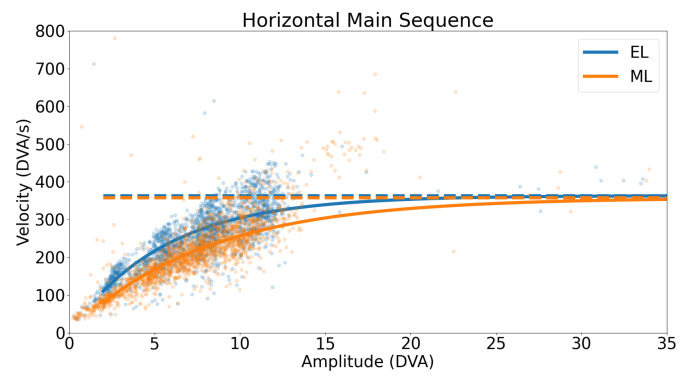
Horizontal main sequence plot for all participants and trials. The dashed line represents the detected asymptotic maximum velocity.

**Figure 12 sensors-23-07753-f012:**
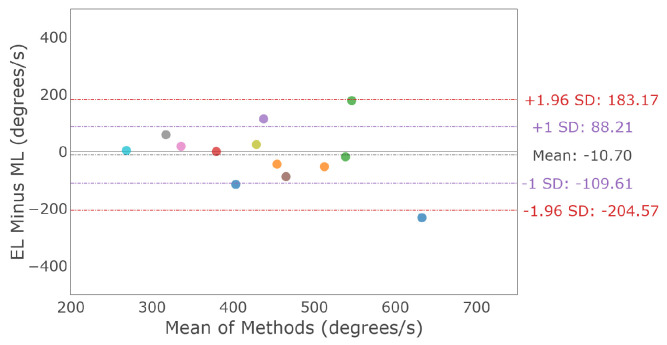
Bland–Altman plot of the detected maximum horizontal saccade velocity between EyeLink and MindLink. Colors distinguish individual participants.

**Figure 13 sensors-23-07753-f013:**
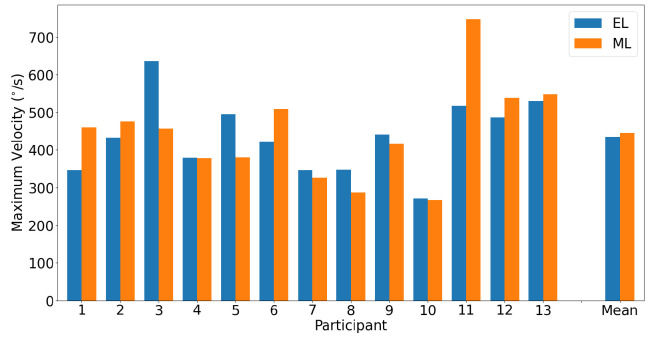
Bar plot of the detected maximum horizontal saccade velocity as measured by both EyeLink and MindLink for each participant, as well as mean values.

**Figure 14 sensors-23-07753-f014:**
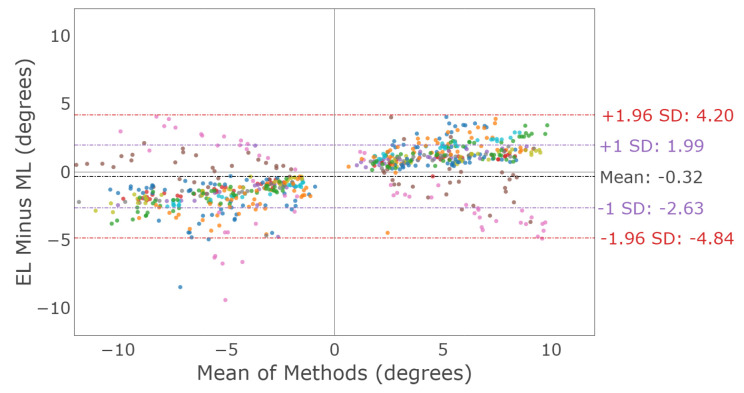
Bland–Altman plot of the vertical saccade amplitude between EyeLink and MindLink. Colors distinguish individual participants.

**Figure 15 sensors-23-07753-f015:**
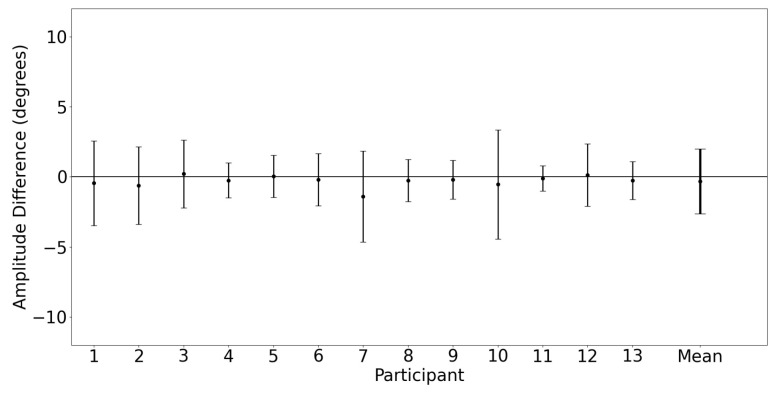
Mean ± SD difference (EyeLink and MindLink) of the vertical saccade amplitude for each participant, as well as the mean ± SD difference averaged across all participants and trials.

**Figure 16 sensors-23-07753-f016:**
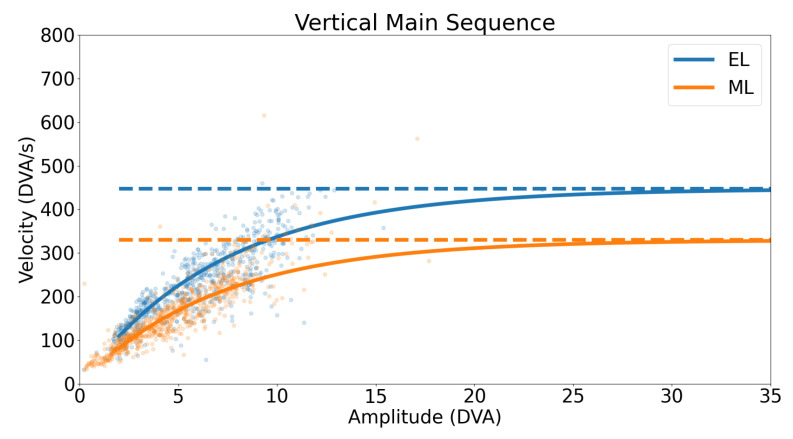
Vertical main sequence plot for all participants and trials. The dashed line represents the detected asymptotic maximum velocity.

**Figure 17 sensors-23-07753-f017:**
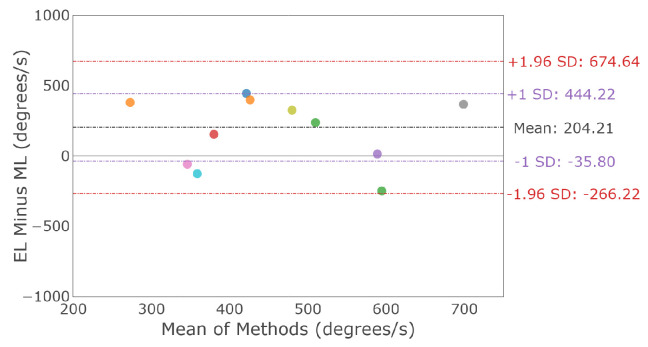
Bland–Altman plot of detected maximum vertical saccade velocity between EyeLink and MindLink. Colors distinguish individual participants.

**Figure 18 sensors-23-07753-f018:**
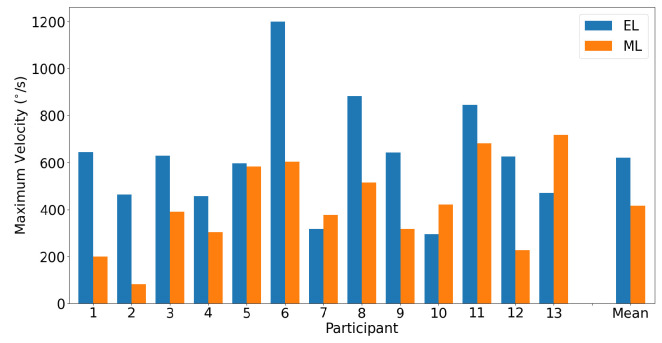
Bar plot of the detected maximum vertical saccade velocity as measured by both EyeLink and MindLink for each participant, as well as mean values.

**Figure 19 sensors-23-07753-f019:**
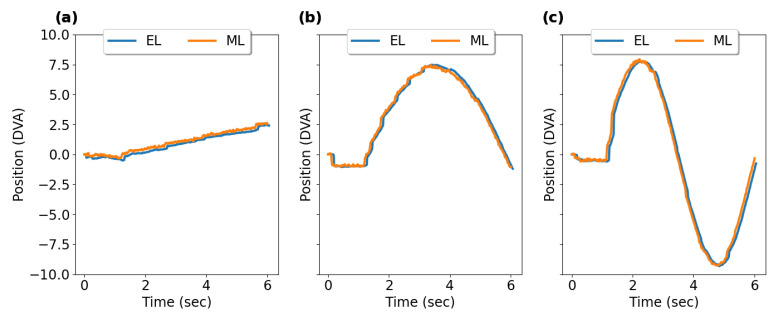
Smooth pursuit eye position profiles from average gaze position data recorded from EyeLink II (EL) and AdHawk MindLink (ML): (**a**) 0.5 $\frac{{}^{\circ }}{s}$, (**b**) 5 $\frac{{}^{\circ }}{s}$, and (**c**) 10 $\frac{{}^{\circ }}{s}$.

**Figure 20 sensors-23-07753-f020:**
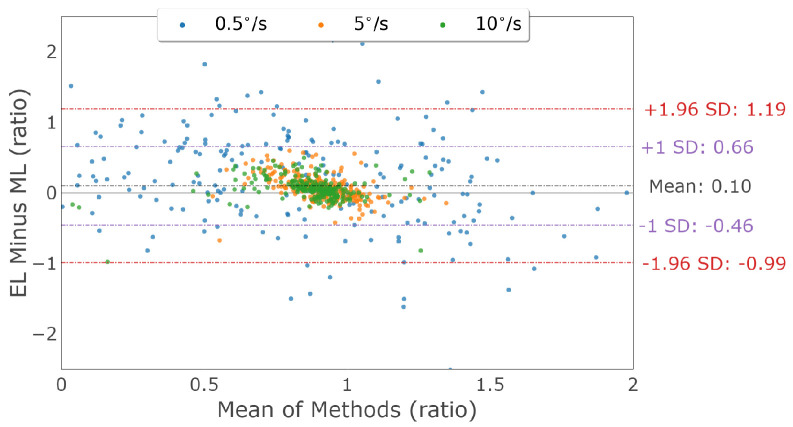
Bland–Altman plot of horizontal velocity gain between EyeLink and MindLink. Colors distinguish smooth pursuit target speeds.

**Figure 21 sensors-23-07753-f021:**
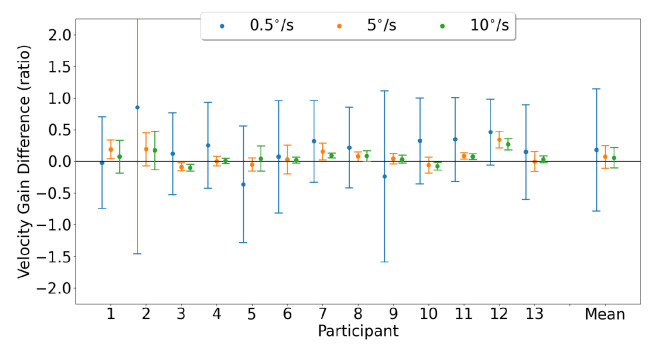
Mean ± SD difference (EyeLink and MindLink) of horizontal velocity gain for each participant, as well as the mean ± SD difference averaged across all participants and trials.

**Figure 22 sensors-23-07753-f022:**
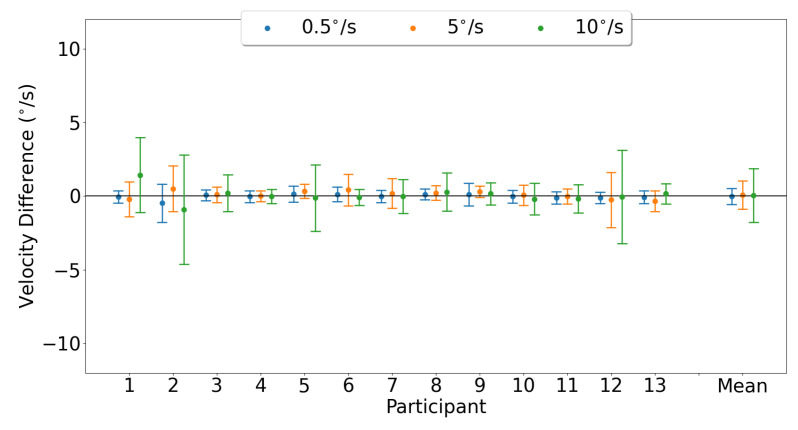
Mean ± SD difference (EyeLink and MindLink) of the horizontal velocity measurement for each participant, as well as the mean ± SD difference averaged across all participants and trials.

**Figure 23 sensors-23-07753-f023:**
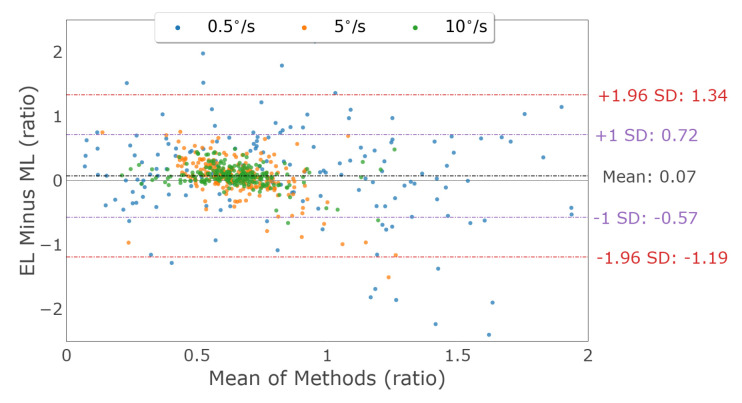
Bland–Altman plot of the vertical velocity gain between EyeLink and MindLink. Colors distinguish smooth pursuit target speeds.

**Figure 24 sensors-23-07753-f024:**
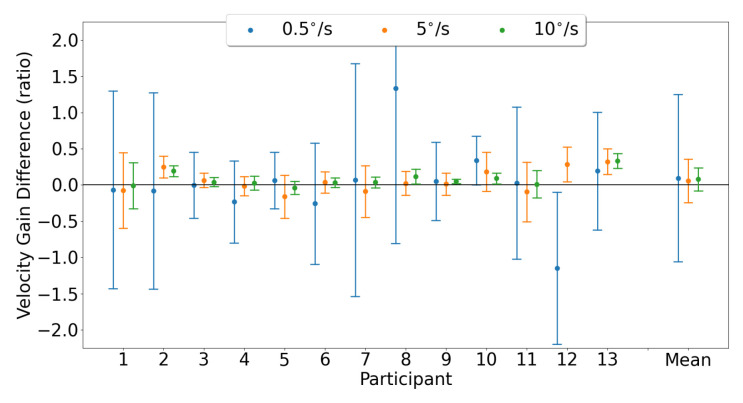
Mean ± SD difference (EyeLink and MindLink) of the vertical velocity gain for each participant, as well as the mean ± SD difference averaged across all participants and trials.

**Figure 25 sensors-23-07753-f025:**
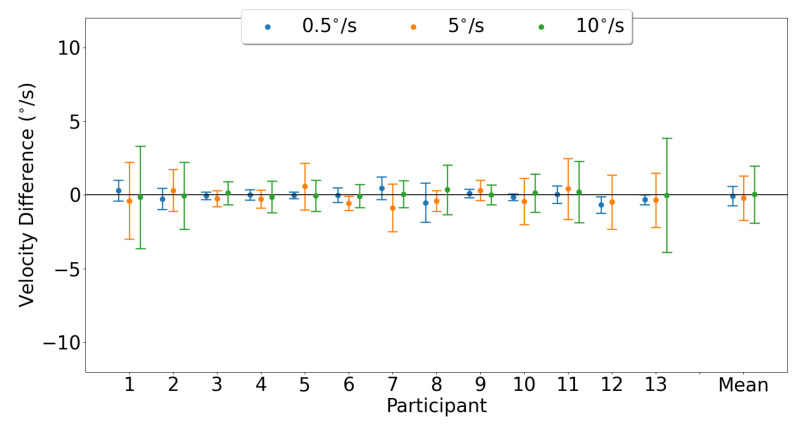
Mean ± SD difference (EyeLink and MindLink) of the vertical velocity measurement for each participant, as well as the mean ± SD difference averaged across all participants and trials.

**Table 1 sensors-23-07753-t001:** Stimulus parameters for the horizontal and vertical saccade and smooth pursuit tasks.

Task		# of Trials	Target Amplitude
Saccades	Horizontal	175	3–25${}^{\circ }$ (2.9, 5.8, 8.6, 11.4, 12.2, 14.4, 17.3, 20.1, 24.5)
	Vertical	60	0.5–17${}^{\circ }$ (0.5, 2.9, 5.7, 6.4, 8.6, 11.9, 17.2)
Smooth Pursuit	Horizontal	30	0.01, 0.1, 0.2 Hz (0.5, 5, 10 $\frac{{}^{\circ }}{s}$)
	Vertical	30	0.01, 0.1, 0.2 Hz (0.5, 5, 10 $\frac{{}^{\circ }}{s}$)

**Table 2 sensors-23-07753-t002:** Summary statistics of fixation stability measures for each eye tracker.

Eye Tracker	Measure	n	Mean (${}^{\circ }$)	SD (${}^{\circ }$)
EyeLink	SD (horizontal)	13	0.13	0.05
	SD (vertical)	13	0.18	0.04
	log${}_{10}$BCEA	13	−0.92	0.21
MindLink	SD (horizontal)	13	0.21	0.11
	SD (vertical)	13	0.25	0.10
	log${}_{10}$BCEA	13	−0.63	0.30

**Table 3 sensors-23-07753-t003:** One-sample *t*-test results comparing differences in the measured saccade amplitudes between eyes for EyeLink and MindLink trackers, for horizontal and vertical saccades.

Saccade Direction	Eye Tracker	Mean of Difference	t-Statistic	*p*-Value	df
Horizontal	EyeLink	−0.34	−4.12	0.002	12
	MindLink	−1.29	−4.09	0.002	12
Vertical	EyeLink	−0.28	−4.04	0.002	12
	MindLink	−1.40	−37.3	0.003	12

**Table 4 sensors-23-07753-t004:** Summary statistics of smooth pursuit measures for each eye tracker.

Eye Tracker	Direction	Measure	n	0.5 $\frac{{}^{\circ }}{s}$		5 $\frac{{}^{\circ }}{s}$		10 $\frac{{}^{\circ }}{s}$	
				Mean	SD	Mean	SD	Mean	SD
EyeLink	Horizontal	Speed	13	0.58	0.20	4.72	0.20	9.85	0.74
		Gain	13	1.00	0.37	0.95	0.04	0.90	0.07
	Vertical	Speed	13	0.42	0.10	3.26	0.29	7.32	1.10
		Gain	13	0.67	0.20	0.65	0.06	0.67	0.10
MindLink	Horizontal	Speed	13	0.52	0.15	4.36	0.65	9.22	1.10
		Gain	13	0.84	0.27	0.87	0.13	0.85	0.10
	Vertical	Speed	13	0.49	0.19	2.86	0.55	6.44	1.24
		Gain	13	0.78	0.35	0.57	0.11	0.59	0.11

**Table 5 sensors-23-07753-t005:** Two-way ANOVA results for horizontal velocity gain.

Effect	DFn	DFd	F	*p*	$\eta_{g}^{2}$
Eye Tracker	1.05	12.64	1.21	0.30	0.01
Pursuit Speed	1.21	14.58	0.77	0.42	0.01
Eye Tracker : Pursuit Speed	1.18	14.13	0.63	0.47	0.03

**Table 6 sensors-23-07753-t006:** Two-way ANOVA results for vertical velocity gain.

Effect	DFn	DFd	F	*p*	$\eta_{g}^{2}$
Eye Tracker	1.05	11.50	0.87	0.38	0.01
Pursuit Speed	1.07	11.74	0.02	0.92	0.0004
Eye Tracker : Pursuit Speed	1.17	12.88	0.85	0.39	0.03

## Data Availability

Data are available upon request.

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
