# Peer review of "Investigation of Camera-Free Eye-Tracking Glasses Compared to a Video-Based System"

_sensors, 2023, doi:10.3390/s23187753_

Round 1
Reviewer 1 Report
1. Participants only 13 persons, is this enough for the following one-sample t-test and ANOVA analysis? Also, why only choosing 13 participants, and 4 males, 9 females?
2. Please give the limitations of the manuscript.
3. Conclusion part should include the main research results, research significance and application value. Also, it is a little short, please rewrite the conclusion.
4. There are only 18 references, please supplement new and good references. Please provide sufficient background and include new relevant references in introduction part.
5. In line 17-19, "Eye movements have been widely used to probe sensorimotor and cognitive mechanisms across a diverse range of applications, including development [1–3], medical diagnostics [4,5], and navigation [6,7]." , which is not a good way to cite references so intensively, especially with small total number of references in the whole manuscript.
6. Please also give the research significance and application value in Abstract.
Minor editing of English language required.
Author Response
Comment 1: Participants only 13 persons, is this enough for the following one-sample t-test and ANOVA analysis? Also, why only choosing 13 participants, and 4 males, 9 females?
Response 1: Our recruitment strategy was to include a sample of convenience. Although the sample is not balanced across the sexes, we don’t anticipate that this would affect our conclusions as we are not aware of any differences between males and females for the eye movement metrics examined in this study. Thirteen participants were tested as this is a representative of many eye movement studies. The protocol used in this study was extensive and testing took ~2 hrs because of the number of trials included for each task. Thus, given the number of trials and participants the study provides sufficient insight into the capability of the Mindlink eye tracker.
Comment 2. Please give the limitations of the manuscript.
Response 2: A new paragraph was added: “This study offers the first investigation of common eye movement metrics recorded using a camera-free eye tracking system. The new system was compared with a research grade video-based eye tracker during a simultaneous recording. While this allowed us to compare the same eye movements recorded with both systems, it also introduced the possibility of interference between devices. To address these issues, pilot testing was conducted prior to the experiment where eye movements were recorded separately by each eye tracker and the recordings were compared to assess the signals from the single vs simultaneous recordings. Results from the pilot study did not reveal any significant interference, nonetheless, signal interference between the devices could have had some small influence on our results. Another limitation of the study is the limited number of tasks used (i.e., fixation, saccade, and smooth pursuit). These tasks have been used widely in eye movement studies; however, the capability of the Mindlink system remains to be examined in other tasks where participants are not constrained by a chin rest and the stimulus is presented in more ecological conditions rather than a computer monitor. Finally, the application of the MEMS technology to eye tracking is quite novel [37]. While the current tracking algorithms provide less accurate recordings when compared to the Eyelink eye tracker, it is likely that as the technology matures the tracking algorithms will also improve.”
Comment 3. Conclusion part should include the main research results, research significance and application value. Also, it is a little short, please rewrite the conclusion.
Response 3: The discussion was restructured to highlight the main results for each task. A separate limitation section was also added.
Comment 4. There are only 18 references, please supplement new and good references. Please provide sufficient background and include new relevant references in introduction part.
Response 4 The introduction section was expanded to highlight the utility of eye tracking to provide insight into human information processing, additional references were added in the revised manuscript.
Comment 5. In line 17-19, "Eye movements have been widely used to probe sensorimotor and cognitive mechanisms across a diverse range of applications, including development [1–3], medical diagnostics [4,5], and navigation [6,7].", which is not a good way to cite references so intensively, especially with small total number of references in the whole manuscript.
Response 5: This section was expanded, and more details were added.
Comment 6. Please also give the research significance and application value in Abstract.
Response 6: A sentence was added to highlight the significance of this research.
Reviewer 2 Report
This paper compares the gaze tracking performance of the AdHawk Mindlink and the Eyelink II by a series of experiments. There is a certain amount of workload. However, I think the objective and innovation of this paper is not outstanding. This paper is more like an experimental report since it does not propose any new methods. The following points should also be well addressed:
1. The schematic diagram of the experimental process is suggested to be added to demonstrate how the experiments were conducted.
2. It is recommended to pay more attention to the format of the equations in the paper to determine whether it should be italicized.
3. The design of the figures in the paper is unreasonable. For example, the vertical axis of Figure 1 is set too large, resulting in the curve not prominent. Figures 2 and 3 are more suitable for presenting in the form of two subfigures. The titles of the subfigures are required. Moreover, the juxtaposition of two figures appears unclear.
4. "log10BCEA" in Table 2, "10" should be a subscript. The authors should carefully check the paper and correct these inconsistencies.
Moderate editing of English language required.
Author Response
Comment 1. The schematic diagram of the experimental process is suggested to be added to demonstrate how the experiments were conducted.
Response 1: A schematic diagram of a possible experimental process (pseudo-randomized) was added (Figure 1).
Comment 2. It is recommended to pay more attention to the format of the equations in the paper to determine whether it should be italicized.
Response 2:
Thank you for bringing these formatting issues to our attention. All equations and expressions in the text are now in accordance with the following italicization conventions:
-
Italicized text: variable names (e.g. x, y), parameters (e.g. V, A, t)
-
Non-italicized text: subscripted variable identifiers (e.g. cm, px, max), mathematical functions (e.g. arctan, log), numbers, units
Comment 3. The design of the figures in the paper is unreasonable. For example, the vertical axis of Figure 1 is set too large, resulting in the curve not prominent. Figures 2 and 3 are more suitable for presenting in the form of two subfigures. The titles of the subfigures are required. Moreover, the juxtaposition of two figures appears unclear.
Response 3: Thank you for bringing these design flaws to our attention. All figures were modified as recommended. Figure 2 and 3, as well as 4 and 5 have been combined to appear as two figures.
Comment 4. "log10BCEA" in Table 2, "10" should be a subscript. The authors should carefully check the paper and correct these inconsistencies.
Response 4: Thank you for bringing these inconsistencies to our attention. The log10 subscript has been corrected.
Reviewer 3 Report
General comments:
The paper describes a comparison between camera-free and camera-based eye tracking systems. The authors concluded that camera-free system has an adequate level of accuracy although it was slightly inferior to the camera-based system. My overall opinion is that the paper should be revised and expanded to be acceptable for publication.
Specific comments:
1. The number of references suggests that the authors may not have extensively explored the topic. Please consider expanding the literature review to provide a more comprehensive theoretical foundation for the research. A more in-depth review of various camera-free systems currently in use within the eye-tracking research area would be of great value to a reader.
2. The authors should use term “camera-based” consistently throughout the entire article, including the title.
3. To make it easier for the reader to understand the methodology, I would recommend describing the experimental procedure in a separate subchapter. Please describe the procedure in precise order - how it started, the order of stimuli presentation, the actions the participant was instructed to perform at each stage, etc.
4. The limitations of the study should be discussed more extensively in Conclusions.
Author Response
Comment 1. The number of references suggests that the authors may not have extensively explored the topic. Please consider expanding the literature review to provide a more comprehensive theoretical foundation for the research. A more in-depth review of various camera-free systems currently in use within the eye-tracking research area would be of great value to a reader.
Response 1: The introduction was expanded by adding a summary of camera-free systems:
“Eye tracking methodologies have evolved greatly over time mainly due to technological and computational advances. Cognolato, Atzori, and Müller (2018) outlined seven different eye tracking methods: 1) electrooculogram (EOG), 2) electromagnetic methods (i.e., scleral search coils), 3) contact lenses, 4) limbus/iris-sclera boundary video-oculography, 5) pupil video-oculography, 6) pupil and corneal reflections video-oculography, and 7) dual Purkinje image corneal reflection video-oculography (see [8] for a complete review of each technique). EOG and search coils are two camera-free eye tracking approaches currently available. Search coils provide the most accurate and precise recordings; however, this approach is the most invasive and restrictive because a copper wire coil is embedded in a contact lens and the recording must be completed while the participant is sitting in an oscillating magnetic field. The voltage recorded by the coils represents the orientation of the eyes in the magnetic field. On the other hand, EOG is less invasive because the signal, generated by changes in the electrical potential between the cornea and the retina when the eyes move, is recorded by electrodes placed on the outer canthi of the eyes. The main limitation of the EOG approach is lower accuracy in comparison to other systems [25].”
Comment 2. The authors should use term “camera-based” consistently throughout the entire article, including the title.
Response 2: The term “video-based" has now been adopted across the entire article.
Comment 3. To make it easier for the reader to understand the methodology, I would recommend describing the experimental procedure in a separate subchapter. Please describe the procedure in precise order - how it started, the order of stimuli presentation, the actions the participant was instructed to perform at each stage, etc.
Response 3: Thank you for your recommendation; the experimental procedure has been expanded to define the order and tasks participants were asked to complete in a clearer fashion.
Comment 4. The limitations of the study should be discussed more extensively in Conclusions.
Response 4: A new paragraph was added: “This study offers the first investigation of common eye movement metrics recorded using a camera-free eye tracking system. The new system was compared with a research grade video-based eye tracker during a simultaneous recording. While this allowed us to compare the same eye movements recorded with both systems, it also introduced the possibility of interference between devices. To address these issues, pilot testing was conducted prior to the experiment where eye movements were recorded separately by each eye tracker and the recordings were compared to assess the signals from the single vs simultaneous recordings. Results from the pilot study did not reveal any significant interference, nonetheless, signal interference between the devices could have had some small influence on our results. Another limitation of the study is the limited number of tasks used (i.e., fixation, saccade, and smooth pursuit). These tasks have been used widely in eye movement studies; however, the capability of the Mindlink system remains to be examined in other tasks where participants are not constrained by a chin rest and the stimulus is presented in more ecological conditions rather than a computer monitor. Finally, the application of the MEMS technology to eye tracking is quite novel [37]. While the current tracking algorithms provide less accurate recordings when compared to the Eyelink eye tracker, it is likely that as the technology matures the tracking algorithms will also improve.”
Round 2
Reviewer 1 Report
The authors have addressed all the issues I raised.
Author Response
Comment #1: The authors have addressed all the issues I raised.
Response #1: Thank you for your comments and suggestions.
Reviewer 2 Report
I think a photo of a participant conducting the experiment using the actual system should be placed in the experimental section. Although it is necessary to provide an example of experimental task orders, it is still not clear how participants conducted the experiment. Readers may be interested in the practical operation process. In addition, the layout of the charts is not satisfactory. If two figures are to be arranged side by side, they should be of the same size, as varying heights can greatly affect their aesthetics.
Minor editing of English language required
Author Response
Comment #1: I think a photo of a participant conducting the experiment using the actual system should be placed in the experimental section. Although it is necessary to provide an example of experimental task orders, it is still not clear how participants conducted the experiment. Readers may be interested in the practical operation process.
Response #1: Thank you for your suggestion. A figure (Figure 1) was added to demonstrate the instrumentation, setup and calibration process for the experiment.
Comment #2: In addition, the layout of the charts is not satisfactory. If two figures are to be arranged side by side, they should be of the same size, as varying heights can greatly affect their aesthetics.
Response #2: Thank you for pointing out the necessary aesthetic revisions. All such pairs of figures (6/7, 10/11, 12/13, 14/15, 17/18, 20/21, 23/24) have had their sizing matched so that they are appropriately aligned vertically.